# In-Stream Marine Litter Collection Device Location Determination Using Bayesian Network

**DOI:** 10.3390/su14106147

**Published:** 2022-05-18

**Authors:** Abdullah Battawi, Ellie Mallon, Anthony Vedral, Eric Sparks, Junfeng Ma, Mohammad Marufuzzaman

**Affiliations:** 1Industrial and System Engineering Department, Mississippi State University, Starkville, MS 39762, USA; ahb264@msstate.edu (A.B.); maruf@ise.msstate.edu (M.M.); 2Osprey Initiative, LLC, Mobile, AL 36606, USA; erm385@msstate.edu; 3Coastal Research and Extension Center, Mississippi State University, Biloxi, MS 39532, USA; anthony.vedral@msstate.edu (A.V.); eric.sparks@msstate.edu (E.S.); 4Mississippi-Alabama Sea Grant Consortium, Ocean Springs, MS 39564, USA

**Keywords:** marine debris, marine litter, Litter Gitter, site selection, coastal, decision network, prevention

## Abstract

Increased generation of waste, production of plastics, and poor environmental stewardship has led to an increase in floating litter. Significant efforts have been dedicated to mitigating this globally relevant issue. Depending on the location of floating litter, removal methods would vary, but usually include manual cleanups by volunteers or workers, use of heavy machinery to rake or sweep litter off beaches or roads, or passive litter collection traps. In the open ocean or streams, a common passive technique is to use booms and a collection receptacle to trap floating litter. These passive traps are usually installed to intercept floating litter; however, identifying the appropriate locations for installing these collection devices is still not fully investigated. We utilized four common criteria and fifteen sub-criteria to determine the most appropriate setup location for an in-stream collection device (Litter Gitter—Osprey Initiative, LLC, Mobile, AL, USA). Bayesian Network technology was applied to analyze these criteria comprehensively. A case study composed of multiple sites across the U.S. Gulf of Mexico Coast was used to validate the proposed approach, and propagation and sensitivity analyses were used to evaluate performance. The results show that the fifteen summarized criteria combined with the Bayesian Network approach could aid location selection and have practical potential for in-stream litter collection devices in coastal areas.

## 1. Introduction

Economic development and rapid population growth have led to increasing volumes of waste generation, particularly trash and litter from human daily life and manufacturing activities [1]. Much of this waste eventually enters waterways or water bodies, becoming *floating litter*. Marine debris (often referred to as marine litter), generally synonymous with *floating litter*/*marine litter*, leads to the deaths of many marine organisms [2,3] and millions of dollars of economic losses each year [4,5,6]. Marine litter is an ever-increasing problem with the continuous growth of solid waste generation domestically and globally [7]. Sources of marine litter are ocean-based (e.g., from fishing vessels, stationary platforms, cargo ships, or other vessels) and land-based (e.g., from stormwater discharges, wind, extreme natural events, and waterfront areas such as beaches, piers, harbors, riverbanks, marinas, and docks) [8]. Litter can be found both floating at the surface and sinking to the ocean bed [9]. Among the marine litter distributed worldwide, approximately 82% of marine litter originates from land-based sources [10]. Approximately 275 million metric tons of plastic waste were generated by 192 coastal countries in 2010, with 4.8 to 12.7 million metric tons entering the ocean [7]. Floating litter, in particular, is harmful to the environment, marine life, human health, and the economy. For instance, piles of litter on the shore decrease the aesthetic value of such areas and make them less attractive to local residents and tourists; moreover, ingestion and entanglement caused by marine litter are fatal to marine organisms [11,12,13], and 17% of marine species ingesting/entangled in the litter are listed as near-threatened, vulnerable, endangered, or critically endangered [14]. Non-native species carried by marine litter drifting worldwide pose a major threat to local marine life [15]. For these reasons, there is an urgent need to remove or reduce floating litter in coastal areas to protect and enhance coastal resilience.

In the study area of this research (Northern U.S. Gulf Coast), the primary litter commonly found is plastic, particularly single-use plastic, originating from land-based litter [16]. According to a recent marine litter study, less than 10% of local residents in the Mississippi Gulf Coast region would prefer to visit Mississippi or Louisiana beaches; 54% of beach visitors complain about the water and shoreline quality [17]. Depending on its location, there are generally two ways to remove macro-floating litter: (1) litter in the ocean, using collection vessels/tools to collect; and (2) litter in coastal streams and rivers, using collection traps [15,18,19,20]. Compared with litter in the ocean, transitory stream and river litter are easier to remove as their trajectory is within a predetermined path and can be collected using stationary traps. There are several different in-stream litter collection devices available to rent or purchase that have been used extensively in inland streams; however, most operate similarly. Each typically includes floating booms that guide or collect floating litter, with some containing a centralized receptacle. Although floating in-stream litter collection devices are effective tools, it is necessary to place them systematically in order to yield the most benefit and avoid inefficiencies in collection capabilities due to a lack of systematic installation. Therefore, the selection of installation locations for such devices is critical. Identifying optimal locations for these devices involves multiple factors pertaining to stream hydrology and cost. Given the complexity and multiple siting consideration factors for an in-stream litter collection device, this process can be considered a Multiple Criteria Decision Making (MCDM) problem.

A wide set of MCDM methods, including Analytic Hierarchy Process (AHP), weighted sum approach (SW), multi-attribute value function theory (MAVT), multi-attribute utility function theory (MAUT), analytic network process (ANP), elimination and choice expressing reality (ELECTRE), and the TOPSIS method, can be used to determine the most appropriate location(s). Generally, in most multi-criteria problems, there is no optimal solution that can satisfy all the criteria at the same time; therefore, compromise solutions must be found [21]. MCDM has been used in many selection-related applications, such as supplier selection and order allocation (e.g., [22]), transportation systems [23], material selection (e.g., [24]), employee recruiting (e.g., [25]), sustainable project portfolio selection (e.g., [26]), and manufacturing (e.g., [27]). Among these approaches, TOPSIS, presented by Hwang and Yoon (1981), has become one of the most widely accepted MCDM approaches [28]. TOPSIS enables decision makers to decide among a group of key parameters that maximize the ability to satisfy the stakeholders [21].

Location selection problems involve variability and subjectivity, which require the understanding of overall available information via space and time scales. A statistical modeling approach to handle uncertainty and make detailed, rational, and transparent contingency plans before taking action is needed. One of the most popular methods for integrating this complexity into tangible actions is Bayesian Network (BN). BN decision making is a widely used tool in location selection applications, such as selecting the most sustainable and economical charging stations for electric vehicles [29]. In Singapore, BN is used to decide bridge location to help the land transport authority properly select and optimize optimal bridge locations [30]. A study in southeastern Australia implemented BN theory in a wildfire location selection problem to choose fire station locations with the least cost impact [31]. BN has also been used to facilitate optimal blood logistics network decisions with the consideration of natural disasters [32]. More recently, BN was utilized to evaluate whether the industry needs to adapt additive manufacturing and model and assess the sustainability performance of supply chain networks [33,34]. Given the potential of the BN approach, it holds the capability to be applied to the decision-making process associated with siting in-stream litter collection devices.

In light of the current state of the art, the major contributions of this study over the existing literature are as follows:This is the first study to methodologically identify and prioritize in-stream litter collection device installation sites.A BN approach is proposed to determine suitable in-stream litter collection device installation sites based on four criteria and fifteen sub-criteria identified in this study.Litter collection device locations across the Northern U.S. Gulf Coast have been used to validate the proposed approach.

## 2. Problem Description and Methodological Framework

### 2.1. Tested In-Stream Litter Collection Device—Litter Gitter

The device used in this study is the Litter Gitter, an innovative device for in-stream litter collection with similar characteristics to other comparable devices used globally, such as The Bandalong Bandit, The Water Goat, Trash Trout, and Sungai Watch’s floating Barriers. The Litter Gitter (LG) is a small in-stream collection device developed by Osprey Initiative, LLC, and is designed to intercept floating litter from stormwater runoff. It includes floating booms that use the current to guide trash to a large wire-mesh collection container (shown in Figure 1). The boom system does not have any nets or barriers that suspend through the water column; thus, limited harm will be made to fish and other wildlife. Litter Gitters have been used widely (43 currently deployed throughout the U.S.) in inland streams to capture floating litter.

### 2.2. Bayesian Network (BN)

Bayesian Network (BN), also referred to as a belief network, is utilized for risk assessment and decision making. BN is a probabilistic model built by an expert based on the theory of Bayes. BN is a useful and efficient approach for calculating the prior probability distribution of undiscovered variables that depend on prior observed variables. A BN model, also called a directed graph, involves two major entities: nodes indicating variables and arrows indicating the interrelationship between nodes. Nodes in BN can be categorized into three classes: (i) parent nodes that do not depend on prior nodes; (ii) child nodes that depend on prior nodes (also referred to as their parent nods); and (iii) intermediate nodes that have both parent and child nodes. In addition, every node in BN has a table referred to as a node probability table (NPT). The base probability of a set of variables can be reconstructed if BN has a different set of evidence. Arrows in a BN denote the connections among nodes, and it can be explained by the conditional probability distribution provided by expert knowledge [35].

Through these relationships, experts can use inference on the random variables in the graph via directed arrows. BN is a distinctive tool for calculating new variable probability distribution computations with unknown conditional spotted variables. With BN, both quantitative and qualitative data can be utilized and added to the model for conditional probability calculation. The constructed nodes can take Boolean (yes/no), integer, qualitative (high/medium/low), discrete, or continuous values. BN has the capability to work with nodes of different types, which is considered as one of the main advantages of using this method. The collected data for the nodes are assembled from historical data and expert standpoints [29]. In this study, BN supports experts and decision makers to evaluate all possible options to locate Litter Gitter sites.

Figure 2 illustrates the BN model with six nodes: *N*_1_, *N*_2_, *N*_3_, *N*_4_, *N*_5_, and *N*_6_, where *N*_1_, *N*_2_, and *N*_3_ are parent nodes. They are initial nodes, so they do not depend on the prior variables, while *N*_4_ and *N*_5_ are intermediate nodes. *N*_4_ depends on *N*_1_, and *N*_5_ depends on *N*_2_ and *N*_3_. *N*_6_ is a child or leaf node, and it depends on both *N*_4_ and *N*_5_. Observation reveals the arrow coming out *N*_1_ to *N*_4_, which indicates that *N*_1_ is an independent node, while *N*_4_ depends on *N*_1_. Equation (1) represents a comprehensive full joint probability distribution of a BN involving *n* variables: *N*_1_, …, *N_n_*.
$$P\left(N_{1},N_{2},,N_{3}\ldots .N_{n}\right)=P\left(N_{1}|N_{2},\ldots .N_{n}\right)P\left(N_{2}|N_{3},\ldots .N_{n}\right)P\left(N_{3}|N_{4},\ldots .N_{n}\right)\ldots$$
(1)$$P\left(N_{n-1}|\;N_{n}\right)\;P\left(N_{n}\right)=\overset{n}{\underset{i=1}{\prod }}P(N_{i}|N_{i+1},\ldots .,N_{n})$$

The six variables shown in Figure 2, *N*_1_, *N*_2_, *N*_3_, *N*_4_, *N*_5_, and *N*_6_ in Equation (1) can be simplified because the primary node of each node is known. For example, we recognize that $N_{4}\;$ has exactly one primary node, *N*_1_. Thus, the joint probability distribution of $P\left(N_{1},\ldots .\;N_{n}\right)\;$ can be replaced with $P\left(N_{4}|\;N_{1}\right)$, given that only *N*_1_ has a significant contribution to the existence of *N*_4_. The balanced joint probability distribution variables are delivered in Equation (2).
(2)$$P\left(N_{1},N_{2},N_{3}\ldots ,N_{6}\right)=P\left(N_{1}\right)P\left(N_{2}\right)\;P\left(N_{3}\right)P\left(N_{4}|N_{1}\right)P\left(N_{5}|N_{2},N_{3}\right)P\left(N_{6}|N_{4},N_{5}\right)$$

In Equation (2), we show the first requirement, which is the calculation of the unconditional probability of $P\left(N_{1}\right),P\left(N_{2}\right),\;\mathrm{and}\;P\left(N_{3}\right)$ and then the conditional probability of $P\left(N_{4}|N_{1}\right)P\left(N_{5}|N_{2},N_{3}\right),\;\mathrm{and}\;P\left(N_{6}|N_{4},N_{5}\right)$ to express the joint distribution of $P\left(N_{1},N_{2},N_{3}\ldots ,N_{6}\right)$.

BN is able to update propagation belief or marginal probabilities. Propagation belief can be added to *P(Ni*) after witnessing another node’s performance by observing other variables. The observed variables are referred to as evidence. For example, the conditional probability for variable *N*_6_ given evidence *e*, (e = {*N*_1_, *N*_2_, *N*_3_, *N*_4_, *N*_5_, *N*_6_}), can be calculated as follows [36]:(3)$$P(N_{4}|e)=\frac{P\left(N_{1},\;N_{2},\;N_{3},\;N_{4},\;N_{5},\;N_{6}\right)}{P\left(N_{1},\;N_{2},\;N_{3},\;N_{5},\;N_{6}\right)}=\frac{P\left(N_{1},\;N_{2},\;N_{3},\;N_{4},\;N_{5},\;N_{6}\right)}{\sum_{N_{4}}\left(N_{1},\;N_{2},\;N_{3},\;N_{5},\;N_{6}\right)}$$

The comprehensive conditional probability, represented in Equation (3), can be computed more precisely by discovering conditional self-sufficiency, as mentioned in Equation (4).
(4)$$P(N_{6}|e)=\frac{P(N_{4}|N_{1})P(N_{5}|N_{2},\;N_{3})\;P(N_{6}|N_{4},\;N_{5})}{\sum_{N_{6}}P(N_{4}|N_{1})P(N_{5}|N_{2},\;N_{3})P(N_{6}|N_{4},\;N_{5})}$$

### 2.3. Conjoint Criteria Utilized for Assessing Litter Gitter Site Selection

Criteria assessment plays a significant role in the site selection of an LG with the continuous growth of solid waste generated in water-based environments. Therefore, in this study, the criteria assessment of LG contributing to the site selection focuses on technical, economical, and environmental perspectives. These perspectives are considered to ensure the suitability and safety of the LG and crew members. The sub-criteria connected with suitability and technical criteria were determined by the following procedure. Firstly, the academic literature and feasibility research studies related to marine litter were collected and evaluated, and the initial sub-criteria were constructed accordingly. Secondly, the expert opinions were merged in the scopes of marine litter. Lastly, the less critical sub-criteria were cast off. Figure 3 illustrates the criteria and sub-criteria considered for site selection of the Litter Gitter. The details of the sub-criteria are addressed below.

#### 2.3.1. Stream Characteristic

Seven sub-criteria, namely, flow rate reduction, bank steepness, bank composition, linear, navigability, creek width, and hydrologic flashness, are considered for the stream characteristic criteria. These criteria were developed during numerous interviews with the owner of Osprey Initiative, LLC, who developed Litter Gitter. The noted criteria come from first-hand experience of installing the Litter Gitters in a variety of environments and geographical locations.

Flow Rate Reduction: The LG should be placed downstream of a drop. A drop in power usually occurs when a stream straightens out, is downstream of a significant elevation change (i.e., a waterfall), or widens out. This will allow the water to flow smoothly and booms to sit correctly in the water.Bank Steepness: When banks are too steep, the booms attached to the LG do not lay correctly and can cause gaps that allow trash to bypass the trap. This gap occurs near the edge of the bank.Bank Composition: The preferred method for securing an LG is to use a tree on either side of the stream. If a tree is not available, metal t-stakes could be used.Linear: Traps should be placed in the straightest portion of the stream. Putting the trap in a turn/curve could cause the water to flow nonlinearly and allow trash to accumulate on the sides of the LG, leading to escape.Navigability: This sub-criterion refers to navigable waterways. These waterways are used for ship movement. Hence, navigable waterways are not appropriate for LG. Navigable waters that are found in the U.S. refer to waters that are subject to tidal flow, and may be used, are reported as used in the past, or may in the future be used for transport that is either interstate or foreign commerce [37].Creek Width: Streams between 20 and 40 ft are best suited for LG. Larger streams tend to have high flow capacity, which puts a strain on the boom system used to anchor the LG. Additionally, larger streams can carry natural debris items such as logs or trees. These large debris items can put more tension on the boom system, causing them to break free from the LG, thus causing the trap to malfunction.Hydrologic Flashness: Flashness refers to the frequency with which rapid, short-term changes in streamflow occur, especially during events where there is runoff and significant rain. An ideal LG placement would be in a stream that does not have significant flashes (10 ft or less). Sudden changes in water flow can cause extra tension to be placed on the anchor points for the LG.

#### 2.3.2. Upstream Characteristics

The three sub-criteria considered for the upstream characteristics are impervious surfaces, population density, and major road crossings. The following criteria are also based on interviews with staff from Osprey Initiative, LLC, and their field experience. There is a lack of data and literature surrounding sources of upstream litter and more research is needed to support these claims.

Impervious Surfaces: Can cause runoff, which can carry trash into stream systems. Ideally, the LG could be placed downstream of an area that will have high impervious surfaces. Examples include placing the trap downstream of a shopping center rather than upstream before the shopping center. Successful sites for LG may be placed within 0.25 miles of high-intensity developed areas.Population Density: Places with high population density are likely to generate more trash simply because more people are there. Ideally, the LG will be located downstream of an area with high population density.Major Road Crossings: A considerable amount of littering occurs around major road crossings. Ideally, the LG should be placed downstream of the road crossings to collect the litter coming from these road crossings.

#### 2.3.3. Permissions and Permitting

In order to install an LG, a set of permissions or permits are typically required. These permissions could include the U.S. Army Corps of Engineers, city, county, or private property owners. During pre-site selection visits, assessments of the likelihood of receiving permission or permits are considered. These considerations include noting the presence of endangered species or habitats that are sensitive to disturbances, such as nesting grounds. Additionally, the jurisdiction in which the site falls under must be investigated in order to infer the practicality of receiving the appropriate permits within the project timeline.

#### 2.3.4. Hazards

Site safety is important to provide the crew with a secure environment to install and maintain the LGs. A site associated with potential risks is subject to exclusion. On the contrary, the safest site has more priority for selection. If the site receives a high risk rating, the trap cannot be placed at this location. Some examples of things that would increase a sites hazard rating could include a steep entrance to the creek, dangerous parking options, and the continual presence of dangerous animals.

### 2.4. The Bayesian Network (BN) Methodological Framework

Figure 4 systematically illustrates the BN methodological framework used to evaluate an LG site selection problem, more specific to a coastal application. The framework delineates the steps that could be undertaken to reliably validate the LG site selection decisions. Essentially, the framework is classified into the following three stages:**Problem definition and systematic study**: Systematically identifies the necessary criteria and sub-criteria required to site an LG for collecting marine litter in a coastal area. Professional input, accessible literature, and elementary LG installation procedures, as instructed by the Environmental Protection Agency (EPA), the U.S. Army Corps of Engineers (USACE), and the National Oceanic Atmospheric Administration (NOAA) [3,4], were utilized to create the criteria and sub-criteria. Overall, four criteria and fifteen sub-criteria are identified to evaluate a possible site for installing the LG (see Figure 3).**The preliminary stage**: Includes collecting data, formulating, and modeling stages. A suitable link between the criteria and sub-criteria is constructed. Related data are gathered to build the BN model with the information collected via the first stage, and a BN model is constructed for each potential site.**Evaluating stage**: Used to check the reliability and validity of the LG project. The result of the BN will undergo sensitivity analyses during the evaluating stage. If authenticated, the analyst will select the best LG site(s); if not, the first stage will be revised, and the criteria/sub-criteria selection and data collection processes will be reevaluated. The process will continue until each site is validated correctly via sensitivity analysis.

## 3. Floating Litter Case Study Solution

### 3.1. BN Model for Evaluating Candidate Litter Gitter Sites

More than fifty sites in the U.S. were initially studied. Ten candidate sites in the coastal area in the south of the U.S. were finally chosen based on their potential success to potentially install LGs. Figure 5 visualizes the ten candidate sites. The specific locations of the candidate sites are listed in Table 1. These potential locations have a common characteristic to ensure initial successful scenarios. For instance, permitting or permission criteria from the city governorate are mandatory for installing such a project. The model takes these criteria as a base requirement for the project. This section shows a BN model using Agena Risk (https://www.agenarisk.com, accessed on 31 March 2022) software for evaluating the possible location of the LG.

The proposed model was developed using the BN theory. There are four criteria in the proposed model: (i) stream characteristics, (ii) upstream characteristics, (iii) permitting or permission, and (iv) hazards criteria. Based on a professional’s input and literature review, the priority ranking of these criteria is as follows. Permitting or permission (of city, counties, or municipality) is considered the top priority, as permission is mandatory for the project. If permission cannot be granted, the trap cannot be placed at the suggested location. The stream characteristics are considered the second priority since it covers the technical parts of the LG trap project. Without ensuring the availability of all needed requirements, an LG trap cannot be placed. The stream characteristics sub-criterion covers technical parameters that affect crew safety and are considered essential for placing an LG trap. Among all other criteria, the third priority is the upstream characteristics that cover an LG’s potential trash capacity and economic criteria.

Below, we provide a description of how different variables are modeled and contribute to the Bayesian Network methodology.

#### 3.1.1. Modeling of Stream Characteristic

Stream characteristics include types of variables that contribute to LG technical stability. Table 2 shows how the different variables are modeled under the stream characteristics. An explanation behind modeling the variables is further given in Table 2. Boolean (for binary decisions (e.g., true/false)) or Truncated Normal (TNORM) distribution (continuous values) are used to model the variables of specific nodes of the BN introduced in Figure 3.

#### 3.1.2. Modeling of Upstream Characteristics

Three variables contribute to the upstream characteristics of the LG installation, namely, impervious surfaces, population density, and major road crossings. Table 3 shows how the different variables are modeled under the upstream characteristics. An explanation behind modeling the variables is further given in Table 3.

#### 3.1.3. Modeling of Permissions Approval

For permission modeling, the IF condition is used to ensure at least one approval is obtained from either the corps of engineers, city principle, county principle, or private property owner (see Figure 6).

#### 3.1.4. Modeling of Hazards Criteria

The hazard node was calculated manually based on the information provided from original data provided by Osprey. The selected sites were ranked from low to high based on factors that would impact the safety of the crew while installing and maintaining the LG (see Figure 7).

### 3.2. Probability of Site Selection

Table 4 provides the LG site selection probability of the ten candidate sites in the coastal area near the Gulf of Mexico (see Figure 5 and Table 1 for the details about the location of the candidate sites). We used the methodological framework introduced in Figure 4 to evaluate the site selection probability for all the ten candidate sites. The target node for our BN framework is the probability of the LG site selection, which is conditioned based on a set of problem-specific criteria, such as stream characteristics, upstream characteristics, permitting/permission, and hazard criteria. The first selection that stood out to install an LG from the ten candidate sites was Site #7, which is located in Mobile, AL (see Table 4). The probability of selecting this site is 81.8% (see Figure 8). This site satisfied all the critical installation criteria and other necessary sub-criteria. The second selection site, with a probability of ~76%, is in Mobile City, AL (Site #1). Figure 9 visualizes the BN results for this site. One of the reasons for placing Site #1 as a second candidate LG installation location over Site #7 is probably the size of the population, which is slightly smaller in Site #7 than Site #1. Furthermore, hydrologic flashiness is slightly less in this selection than in the first selection. Similarly, we demonstrated the BN results for the third-, fourth-, and fifth-best locations, which are nearly 73% (Site #8), 72% (Site #6), and 68% (Site #9), respectively (see Figure 10, Figure 11 and Figure 12). Note that the BN results for all the top sites can be compared with the standard BN results shown in Figure A1 in Appendix A1.

### 3.3. Sensitivity Analysis

Sensitivity analysis is a method used to validate the constructed Bayesian Network model that investigates the effect of variables on the target node. Sensitive parameters may significantly affect the results of the target node. Analyzing these parameters may help experts direct their efforts more efficiently to obtain a trustworthy Bayesian Network model.

Validation is utilized to compare the current constructed model to the actual result. In Figure 13 and Figure 14, tornado graphs are used to demonstrate the importance of the nodes in determining the probability of selecting a candidate LG site. The variables in the chart are represented in boxes with two conditions, “true” and “false.” The longer the box, the greater the influence on determining the probability of the candidate LG sites (target node). The tornado graph shows hazards, permitting, upstream characteristics, and stream characteristics criteria with a rough difference of 0.25. The analysis of the tornado chart indicates different influences among all criteria. Therefore, we can say that there is a similar influence on the target node among all criteria.

Hazard in the tornado graph explains that the probability of selecting the first LG location (“true”) starts from 0.47 (when the hazard criteria is “false”) to 0.72 (when a hazard criterion is “true”). The probability of selecting the first litter location is 0.6, given that the hazard criteria is achieved. This range (0.47–0.72) is precisely the bar in the tornado graph explained in Figure 13. For permitting criteria, upstream characteristics, and stream characteristics criteria, the probability of selecting the first LG location is 0.39–0.64, 0.44–0.69, and 0.52–0.77, respectively. 

The chart’s vertical line mainly indicates the marginal probability for the first selection LG location being “true” (0.60). The likelihood of selecting the first selection site location is less sensitive to the changes in our model since all criteria length differences are almost the same between all constructed criteria. Therefore, decision makers must give equal attention to all criteria [29].

## 4. Conclusions

Floating litter is one of the most widespread threats that can negatively impact the quality of life in coastal areas. In this study, we developed a methodological framework to assess optimal locations to install an LG, an example of an in-stream litter collection device that has the capability to reduce the quantities of floating litter in local habitats. We identified four criteria and fifteen sub-criteria to determine the most appropriate location to install an LG. The criteria and sub-criteria were incorporated under the BN framework to quantify the selection probability of a site among a set of candidate sites. The developed BN model combines both qualitative and quantitative input for each potential site location. The Northern Gulf of Mexico Coast in the U.S. was used as a case study to validate the BN framework for installing LGs and similar collection devices. All the candidate sites were assessed based on the consideration of the site’s technical and safety factors. We performed a sensitivity analysis to understand the contribution of each criterion for determining the LG site. We found that the contribution of the criteria is ranked as recommended from an expert team and research studies (hazard, permitting, stream characteristics, and upstream characteristics). However, decision makers must place equal focus on all criteria. The proposed BN decision-making framework and the generated insights have the potential to help stakeholders select the most effective sites for in-stream collection devices such as the LG.

## Figures and Tables

**Figure 1 sustainability-14-06147-f001:**
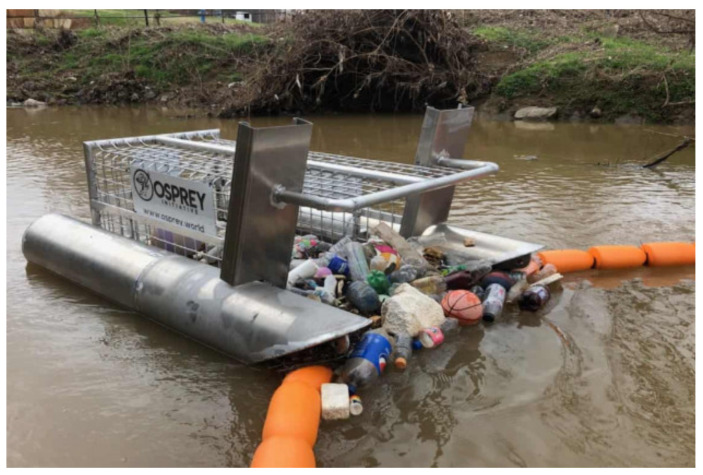
Litter Gitter in Auguste Bayou, Biloxi, Mississippi.

**Figure 2 sustainability-14-06147-f002:**
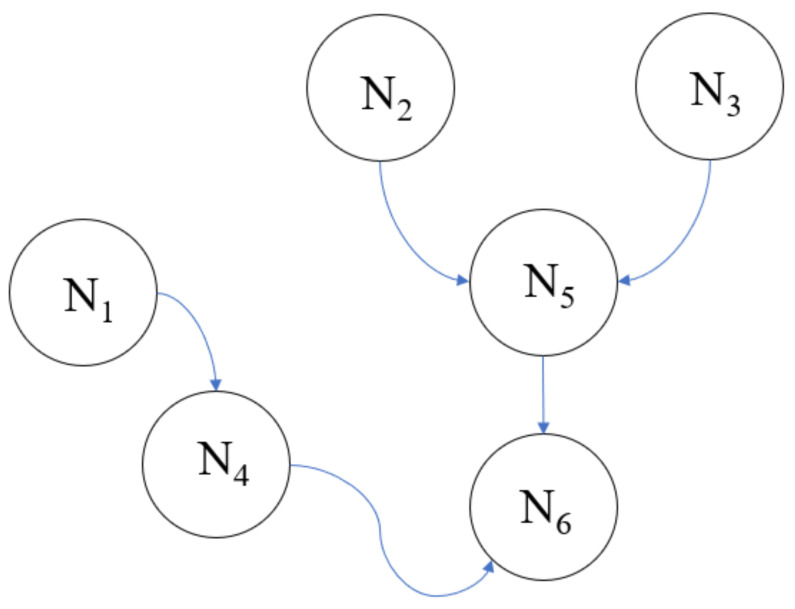
Illustration of Bayesian Network model with six nodes.

**Figure 3 sustainability-14-06147-f003:**
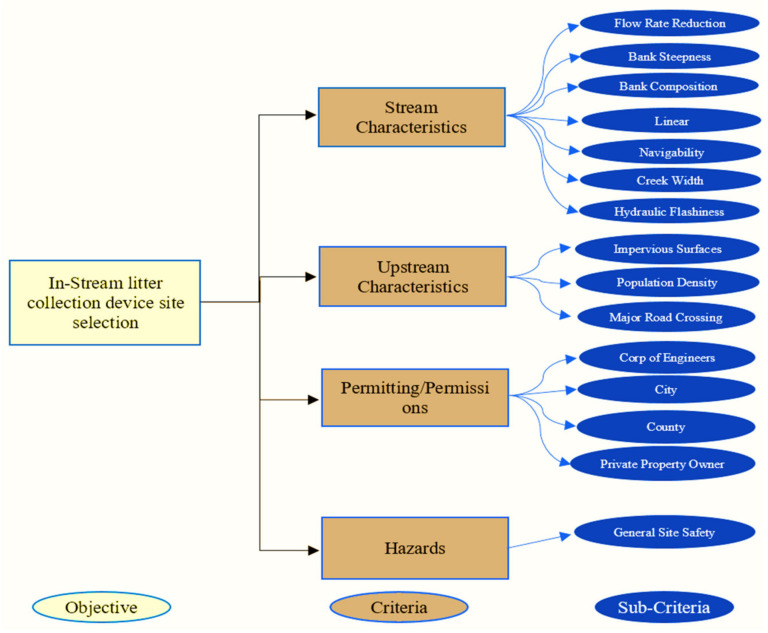
Criteria and sub-criteria for evaluating LG site selection.

**Figure 4 sustainability-14-06147-f004:**
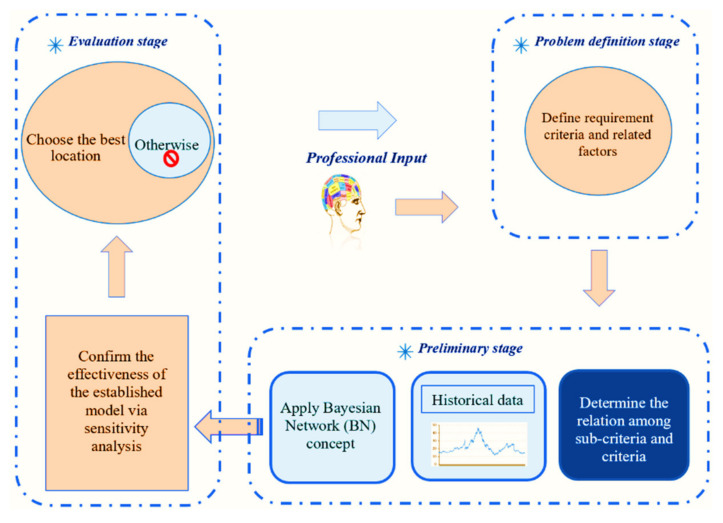
Methodology framework for evaluating an LG site selection problem.

**Figure 5 sustainability-14-06147-f005:**
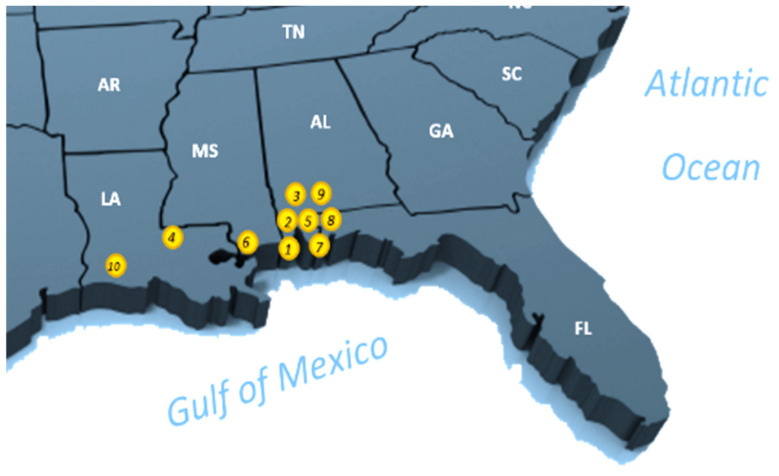
The general geographical locations of the ten candidate LG installation sites.

**Figure 6 sustainability-14-06147-f006:**
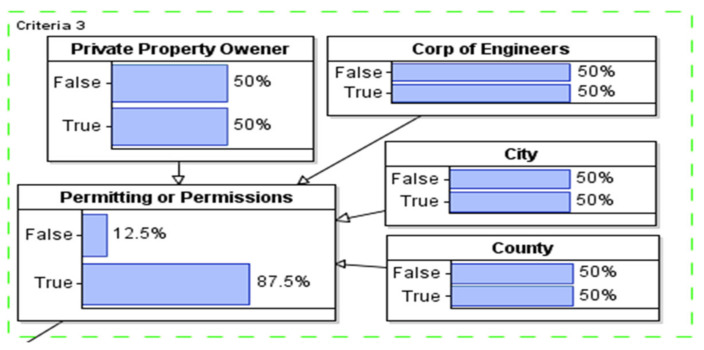
Securing permission modeling from city/county governor.

**Figure 7 sustainability-14-06147-f007:**
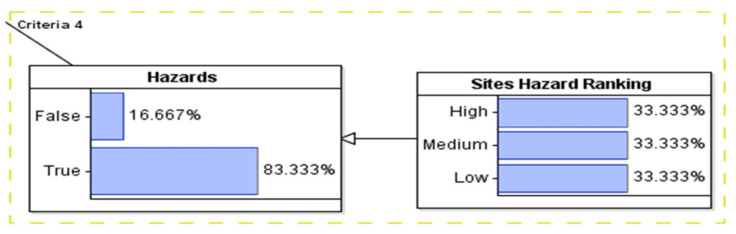
Modeling the hazards variable.

**Figure 8 sustainability-14-06147-f008:**
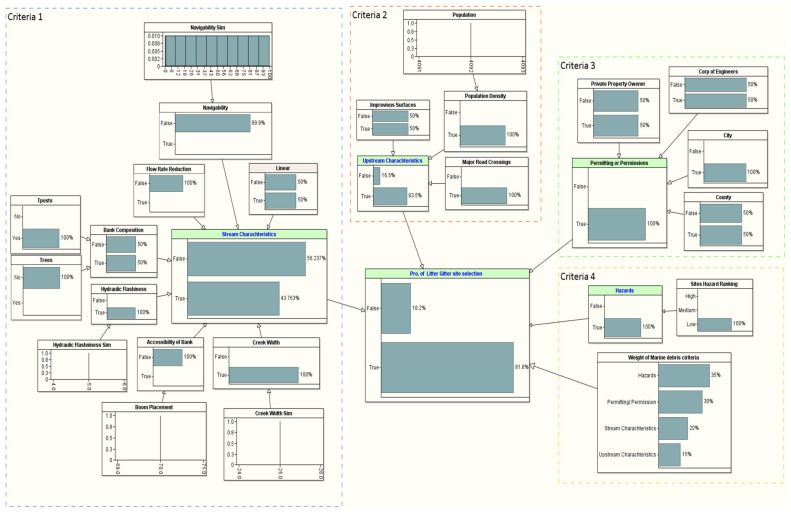
The developed BN model for the first LG selection (Site #7).

**Figure 9 sustainability-14-06147-f009:**
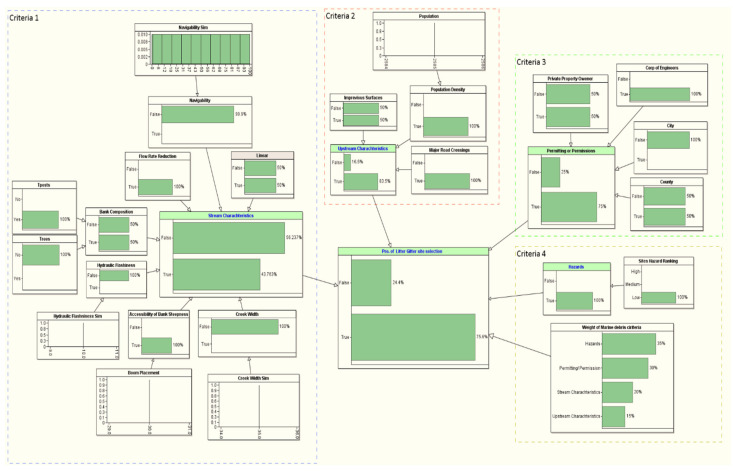
The developed BN model for the second LG selection (Site #1).

**Figure 10 sustainability-14-06147-f010:**
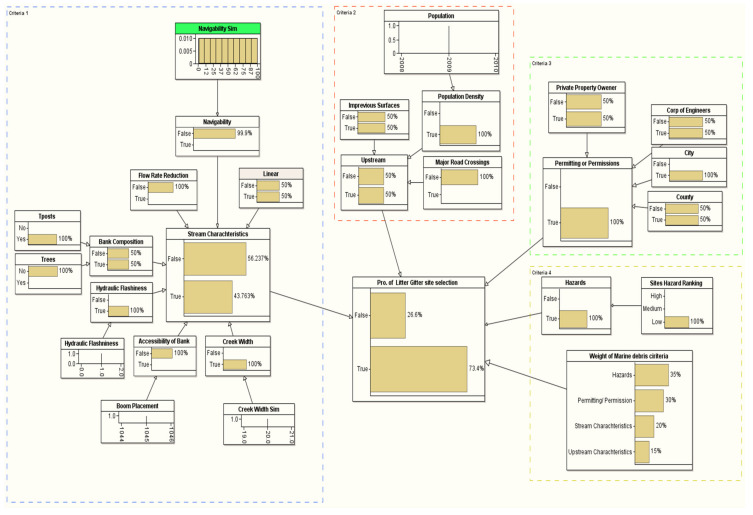
The developed BN model for the third LG selection (Site #8).

**Figure 11 sustainability-14-06147-f011:**
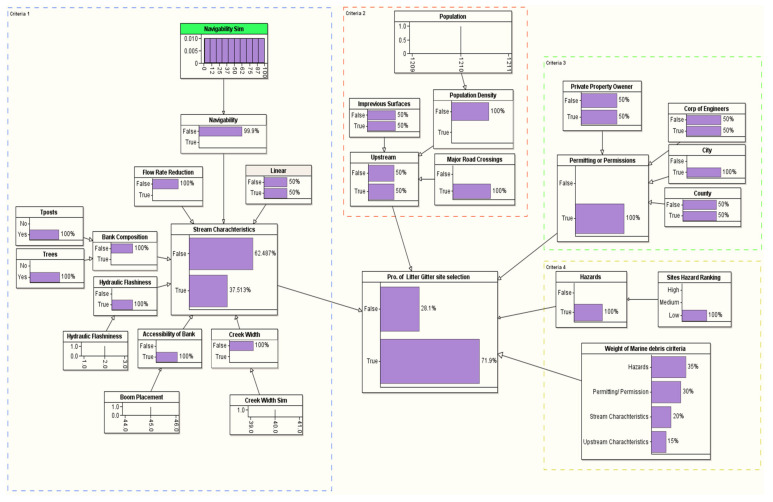
The developed BN model for the fourth LG selection (Site #6).

**Figure 12 sustainability-14-06147-f012:**
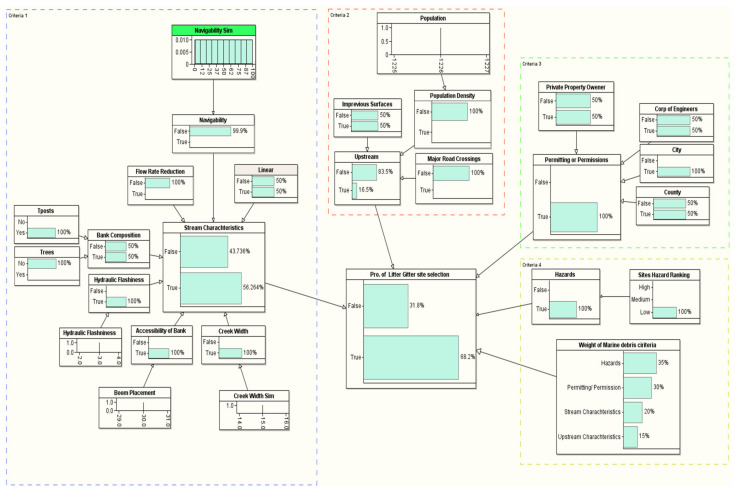
The BN model for the fifth LG selection (Site #9).

**Figure 13 sustainability-14-06147-f013:**
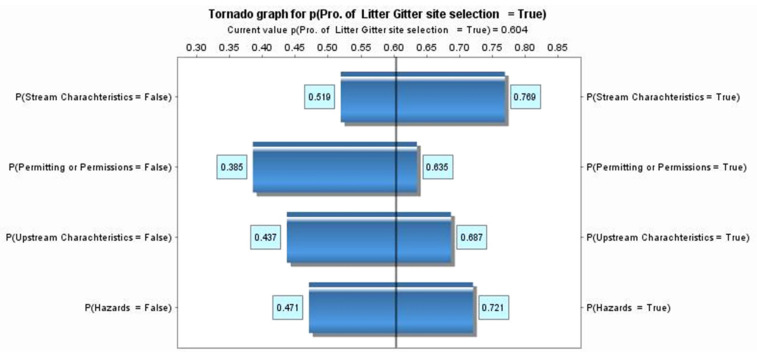
The tornado chart shows the nodes that have the most impact on selecting the first site, “true”.

**Figure 14 sustainability-14-06147-f014:**
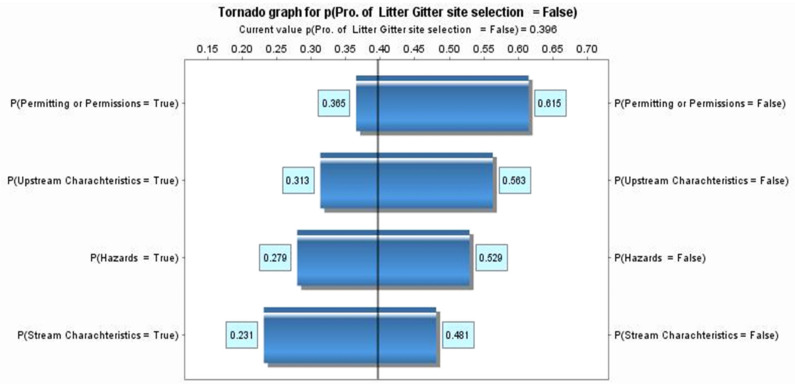
The tornado chart shows the nodes that have the most impact on selecting the first site, “false”.

**Table 1 sustainability-14-06147-t001:** The potential LG sites in the coastal area.

Site	City	State	Location Name	LG Location	
				Latitude	Longitude
1	Mobile	AL	DR-Eslava Sage	30.67321	−88.11316
2	Mobile	AL	3MC-1MC Lawrence	30.70263	−88.05416
3	Daphne	AL ^1^	DO-D’Olive Creek US98	30.65274	−87.91149
4	Ponchatoula	LA ^2^	LP-Ponchatoula Creek_I-55	30.45581	−90.47149
5	Foley	AL	BS-UTBS Cedar	30.38675	−87.69209
6	Biloxi	MS ^3^	BBB-Keegan Bayou_I-110	30.40612	−88.89473
7	Mobile	AL	DR-Montlimar Canal Michael Blvd	30.66329	−88.13669
8	Mobile	AL	3MC-3MC Infirmary	30.69957	−88.07901
9	Mobile	AL	3MC-3MC_Langan Park	30.70562	−88.16482
10	Hammond	LA	LP-Yellow Water River, Adams Rd	30.45864	−90.50564

^1^ Alabama; ^2^ Louisiana; ^3^ Mississippi.

**Table 2 sustainability-14-06147-t002:** Modeling of variables contributing to the stream characteristics.

Variable	Modeling Procedure	Explanation
Flow Rate Reduction	IF (Flow Rate = 1, “True”, “False”)	It is difficult to position the LG in the direction of rapid rivers. High flow will cause the trash to escape the LG. Therefore, the LG needs to be placed in a downstream drop of energy. In the model, 1 represents a stable location, and 0 represents a disturbance location.
Bank Steepness	TNORM (µ = 57, σ2 = 33, LB = 10, UB = 90)	According to the historical data, bank steepness follows a truncated normal distribution with a mean of 57.
Bank Composition	IF (Bank Composition = 1, “True”, “False”)	As described earlier, the bank composition must hold to either a tree or a metal fence t-stakes. If not, the trap cannot be placed.
Linear	IF (Trap Linearity = 1, “True”, “False”)	Linearity is another critical aspect that follows a Boolean distribution. It has an equal probability of finding it or not. The threshold that traps linearity must be equal to one.
Navigability	IF (Navigability = 1, “True”, “False”)	Navigability is an essential aspect of LG installation. The IF condition ensures no navigability in the intended area. The one indicated area has no navigability. The area is calm enough for the trap to be placed.
Creek Width Calculation of Creek Width	TNORM (µ = 35, σ2 = 12, LB = 10, UB = 50) IF (Creek Width < 31, “True”, “False”)	According to the collected data, the creek width follows truncated normal distribution with an average of 35. To ensure the trap’s safe operation, we want to provide less interruption to the LG. Thus, the preferred creek width is less than 31.
Hydrologic Flashness	IF (Hydrologic Flashiness < 9.0, “True”, “False”)	The greatest accepted safe operation of HF is 9 ft.

**Table 3 sustainability-14-06147-t003:** Modeling of variables contributing to the upstream characteristics.

Variable Name	Modeling Procedure	Explanation
Impervious Surfaces	NORM (µ = 0.25, σ2 = 0.03)	Impervious surfaces follow a normal distribution with a mean of 0.25 miles and a variance of 0.03.
Population Density Setup Population Density Calculation	TNORM (µ = 2193, σ2 = 1045, LB = 647, UB = 4160) IF (Population Density Setup > 1800, “True”, “False”)	The population density follows a truncated normal distribution with an average of 2193 and variance of 1045; the lower bound is 647, and the upper bound is 4160. As explained earlier, a site with a high density level would be more favorable since the trap will capture more trash. A site would be more useful if the population density level is more than 1800.
Major Road Crossings	IF (Major Road Crossing > 1, “True”, “False”)	As described earlier, more trash occurs at major road crossings. The IF condition gives sites located near major road crossings more weight than other sites.

**Table 4 sustainability-14-06147-t004:** Site selection probability of the ten candidate sites in the coastal area near the Gulf of Mexico.

Criteria	Sub-Criteria	Site 1	Site 2	Site 3	Site 4	Site 5	Site 6	Site 7	Site 8	Site 9	Site 10
Stream Characteristics	Flow Rate Reduction	Y *	N *	N	Y	N	N	N	N	N	N
	Bake Steepness	30	90	10	90	60	45	70	45	30	30
	Bank Composition	T posts	Trees	Trees	Trees/ Tposts	Trees	Trees/ Tposts	T posts	T posts	T posts	T posts
	Linear	Y	Y	Y	Y	Y	Y	Y	Y	Y	Y
	Navigability	NN	NN	NN	NN	NN	NN	NN	NN	NN	NN
	Creek Width	35 ft	20 ft	50 ft	35 ft	10 ft	40 ft	25 ft	20 ft	15 ft	35 ft
	Hydrologic Flashiness	10 ft	5 ft	3 ft	10 ft	3 ft	2 ft	5 ft	1 ft	3 ft	6 ft
Upstream Characteristics	Impervious Surfaces	Y	Y	Y	Y	Y	Y	Y	Y	Y	Y
	Population Density	2584	1832	1986	983	646	1210	4092	2009	1226	119
	Major Road Crossings	Y	Y	N	Y	N	Y	Y	N	N	N
Permitting/Permissions	Corp of Engineers	Y	Y	N	N	N	Y	Y	Y	Y	N
	City	N	N	Y	N	Y	Y	Y	Y	Y	N
	County	N	N	N	Y	N	N	N	N	N	Y
	Private Property Owner	N	N	N	Y	N	N	N	N	N	N
Hazards	General Site Safety	L *	H *	M *	M	H	L	L	L	L	L
Probability of site selection- True (%)		75.6	50.6	63.4	55.1	43.2	71.9	81.8	73.4	68.2	58.8

* Y—yes; N—no; H—high; M—medium; L—low; NN—non-navigable.

## Data Availability

Not Applicable.

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
