# Peer review of "In-Stream Marine Litter Collection Device Location Determination Using Bayesian Network"

_sustainability, 2022, doi:10.3390/su14106147_

Round 1
Reviewer 1 Report
- This manuscript aims to find locations suitable for placing in-stream marine litter collective devices by using the Bayesian Network approach. This approach is commonly used to solve multi-criteria decision making problems such as the site selection for LG (Litter Gitter). In general, the manuscript is well structured with specific aims, appropriate methodology and outcomes derived. However, some key issues are identified and need to be for further clarification, elaboration or revision.
- Line 190-192. The manuscript goes that the criteria assessment of LG contributed to the site selection will focus on the technical, economical and environmental perspectives. However, none of the criteria and associated sub-criteria (as seen in Fig. 3) for evaluating LG site selection are relating to ‘technical’, and ‘economical’ perspectives. As an illustration, the ‘Technical’ perspective, by its name, refers to the ‘technical details of the LG design’, but none of the selected criteria deals with this perspective.
- Furthermore, the manuscript also goes that these perspectives are considered to ensure the sustainability and safety of LG and crew members. What ‘sustainability’ means in the scope of placing in-stream LG?
- P5-6. As for the criteria - permissions, it is hard to image this factor plays a role in contributing to the ‘suitability’ of the site. Authors have to elaborate more on this point in order to justify this criterion. And, how we can make sure the ‘data’ regarding the ‘at least one approval is obtained from either corps of engineers, city principles etc.’ when we just calculate the probability of selection but not indeed submit the application for permissions?
- P 10. Hazards criteria include three sub-criteria: bank steepness, creek width, hydraulic flashiness. But these sub criteria are also components constituting the criterion – stream characteristic. In other words, hazard criteria can be assimilated into the node of stream characteristic. There is need to clarify or elaborate more on the reasons that hazard has to be separately dealt with as well as the problem of double counting the contribution of bank steepness, creek width and hydraulic flashiness to the probability of site selection.
Author Response
Please see the attached response. Thanks,

Reviewer 2 Report
Journal: Sustainability
The authors investigate the assessment of the probability of site selection at ten locations in the Gulf of Mexico area based on pre-defined criteria and the use of Bayesian Network technology. The manuscript is interesting because it deals with the attempt to implement a barrier or trap for waste that grows exponentially. Regarding the paper, I have no comments on correcting parts of the manuscript. However, the authors did not take into account the feasibility i.e. economic analysis as well as visual impact - it is important to choose the ideal place, but it is important that the device does not aesthetically affect the landscape.
Specific comments:
Line 44: „by 192 coastal counties“ - Clearly define the area of the world.
Line 165: „Fig. 2 illustrates the BN model with six nodes“ - This text should be before Figure 2.
Line 190: The abbreviation LG is mentioned here for the first time, give its explanation
Figure 4 should be found within Chapter 2.4. , no one before.
Line 309: „(https://www.agenarisk.com/)“ - The web link should be added to the list of references and numbered within the text.
The resolution of Figures 3, 4, 6, 7, and in particular 9, 10, 11, 12, 13, 14, 15, 16 is unacceptable. The text cannot be read!
Author Response
Please find attached response to reviewer 2. Thanks,

Round 2
Reviewer 1 Report
The manuscript is well revised and response is handled well to each inquiry. No further comments.